# Inclusion of Soluble Fiber in the Gestation Diet Changes the Gut Microbiota, Affects Plasma Propionate and Odd-Chain Fatty Acids Levels, and Improves Insulin Sensitivity in Sows

**DOI:** 10.3390/ijms21020635

**Published:** 2020-01-18

**Authors:** Chuanhui Xu, Chuanshang Cheng, Xiu Zhang, Jian Peng

**Affiliations:** 1Department of Animal Nutrition and Feed Science, College of Animal Science and Technology, Huazhong Agricultural University, Wuhan 430070, China; xuchuanhui001@webmail.hzau.edu.cn (C.X.); chengcs1989@gmail.com (C.C.); zhangxiu199205@163.com (X.Z.); 2Key Laboratory of Animal Nutrition and Feed Science, Ministry of Agriculture and Rural Affairs, WENS Research Institute (Technology center), Yunfu 527300, China; 3The Cooperative Innovation Centre for Sustainable Pig Production, Wuhan 430070, China

**Keywords:** soluble fiber, propionate, odd-chain fatty acids, insulin sensitivity, sow

## Abstract

The transition from pregnancy to lactation is characterized by a progressive decrease in insulin sensitivity. Propionate increases with dietary fiber consumption and has been shown to improve insulin sensitivity. Recent studies suggest that plasma odd-chain fatty acids [OCFAs; pentadecanoic acid (C15:0) and heptadecanoic acid (C17:0)] that inversely correlated with insulin resistance are synthesized endogenously from gut-derived propionate. The present study investigated the effects of soluble fiber during gestation on gut microbiota, plasma non-esterified fatty acids and insulin sensitivity in sows. Sows were allocated to either control or 2.0% guar gum plus pregelatinized waxy maize starch (SF) dietary treatment during gestation. The SF addition changes the structure and composition of gut microbiota in sows. Genus *Eubacterium* increased by SF addition may promote intestinal propionate production. Moreover, the dietary SF increased circulating levels of plasma OCFAs, especially C17:0. The SF-fed sows had a higher insulin sensitivity and a lower systemic inflammation level during perinatal period. Furthermore, the plasma C15:0 and C17:0 was negatively correlated with the area under curve of plasma glucose after meal and plasma interleukin-6. In conclusion, dietary SF improves insulin sensitivity and alleviates systemic inflammation in perinatal sows, potentially related to its stimulating effect on propionate and OCFAs production.

## 1. Introduction

During normal pregnancy and lactation, the female body undergoes substantial hormonal, immunological, and metabolic changes to support the growth and development of offspring [1,2]. However, the transition from pregnancy to lactation is characterized by physiological and metabolic changes, such as a progressive decrease in insulin sensitivity during late gestation and early lactation [3], which may unfortunately result in decreased lactation feed intake of sows [4]. The feed intake of breeding sows during lactation directly affects the overall productivity of sow operations [5,6]. Our previous studies showed that sows provided with high-fiber diet during gestation significantly increased their voluntary feed intake during lactation [7,8,9], probably as a result of insulin sensitivity improvement. Nevertheless, the mechanism by which dietary soluble fiber exerts the benefit is poorly understood.

Coinciding with the metabolic and immune alterations, there are noticeable changes in fecal microbiota of pregnancy individuals during and after pregnancy [10,11]. These microbiota alterations might also be directly linked with the maternal immune and metabolic profile and thereby contribute to the development of pregnancy complications [10,12], as well as affect the metabolic and immunological health of the offspring [13,14]. As the major energy source for gut microbiota, dietary fiber is believed to have significant effects on the composition and diversity of microbiota [15,16]. Bacterial fermentation of these compounds results in the production of short chain fatty acids (SCFAs), which could decrease fatty acid flux and improve insulin sensitivity in humans [17]. Increased plasma non-esterified fatty acids (NEFAs) link obesity with insulin resistance and type 2 diabetes mellitus [18]. However, in contrast to the even-chain saturated fatty acids (SFA), the odd-chain fatty acids (OCFAs) pentadecanoic acid (C15:0) and heptadecanoic acid (C17:0) account for only a small proportion of long-chain fatty acids in human plasma, but their presence is associated with a lower incidence of ischemic heart disease and type 2 diabetes [19,20] It is assumed that OCFAs are not synthesized endogenously by mammals and therefore reflect dietary habits. However, recent research found that propionate is a direct precursor for OCFA formation in humans and rodent models [21]. Propionate derives mainly from intestinal bacterial fermentation of dietary fibers and is likely the relevant source of circulating OCFAs, especially heptadecanoic acid [21]. On the basis of these results, we hypothesized that OCFAs de novo synthesis also occurs in sows.

Due to the various physicochemical properties of dietary fiber, physiological effects of dietary fiber also vary greatly as reviewed by Hamaker and Tuncil [22]. In our previous studies, we found that propionate in cecal content of rats were increased after pregelatinized waxy maize starch plus guar gum (SF) supplementation [23]. Based on these data, we hypothesize that the inclusion of SF in a gestation diet would modify the intestinal microbiota and that the modifications would be associated with changes in plasma concentrations of propionate and OCFA, insulin sensitivity and inflammatory cytokine of sows. Therefore, the aim of this study was to determine the effects of SF inclusion in the gestation diet on plasma concentrations of propionate and NEFAs and insulin sensitivity and systemic inflammatory cytokine of sows. The influence of SF treatment on the taxonomic profile of the gut microbiota of sows was also investigated by high-throughput sequencing analysis.

## 2. Results

### 2.1. Dietary Soluble Fiber Changes the Gut Microbiota Diversity and Composition of Sows during Perinatal Period

In non-pregnant individuals, dietary soluble fiber has been shown to exert beneficial regulatory effects on gut microbiota [24,25]. However, the regulation of dietary SF on maternal gut microbiota during pregnancy is largely unclear. Thus, the microbiota diversity and composition of fecal samples in sows were detected by deep sequencing of the V3–V4 region of the 16S rRNA genes. To assess fecal microbial community structure, the alpha and beta diversity were calculated (Figure 1). For richness indexes (observed species, Chao1), breeding stage exhibited remarkable effect on it with data on GES109 being much higher than any other stages (Figure 1A,B). There was an interactive effect between breeding stage and dietary SF (*p* < 0.01). A significant increment in richness indexes at LAC3 with SF inclusion was found. The breeding stage also showed dramatic effects on Shannon index with data on LAC3 being much lower than any other stages (Figure 1C). Interactive effect between breeding stage and dietary SF on Shannon index was also detected to suggest the view that SF supplementation could effectively reduce the dramatic changes of alpha diversity induced by breeding stage. Furthermore, beta-diversity was evaluated by the principal coordinate analysis (PCoA) based on weighted UniFrac distance. PCoA clustering revealed that the gut microbiota of sows was least dispersed in GES30 which formed a well-defined cluster but showed obvious segregation in GES109 and LAC3 (Figure 1D). The gut microbiota of LAC3 sows were subjected to the greatest variation since the cluster was the most diffuse shape (Figure 1D). Notably, the gut microbiota showed obvious segregation from CON-fed to SF-fed sows in GES109 and LAC3 with less interindividual variations in SF-fed sows (Figure 1E,F). Taken together, the addition of SF to pregnancy diet effectively prevent the significant changes in gut microbiota diversity of sows induced by pregnancy.

The remarkable changes in gut microbiota diversity of sows led us to assess the composition of the fecal microbiota. The relative abundances at phylum and genus level of all feces across different breeding stages were shown in Figure 2. The top seven phyla were *Firmicutes*, *Bacteroidetes*, *Spirochaetes*, *Proteobacteria*, *Verrucomicrobia*, *Tenericutes*, and *Fusobacteria* (Figure 2A). These seven phyla accounted for above 98% of the reads for all sows. In addition, the top seven genera of gut microbiota in sows were *Eubacterium*, *Prevotella*, *Oscillospira*, *Bacteroides*, *Treponema*, *Clostridium*, *YRC22*, and *Ruminococcus* (Figure 2B). Significant difference of the phyla *Firmicutes*, *Bacteroidetes*, and *Proteobacteria* and genus *Eubacterium* were further analyzed. Dietary SF supplementation increased the relative abundance of *Firmicutes* and *Eubacterium* and decrease the relative abundance of *Bacteroidetes* and *Proteobacteria* (Figure 2C–F). The phyla *Firmicutes* and *Proteobacteria* were significantly affected by breeding stage. The relative abundance of *Firmicutes* was higher in GES30 and GES109 than in LAC3 (Figure 2C). Additionally, the relative abundance of *Proteobacteria* was the highest in LAC3 (Figure 2E). Interactive effect between breeding stage and dietary SF was found for the three phyla. Besides, the relative abundance of genus *Eubacterium* were higher in GES109 than in GES30 (Figure 2F). The microbial composition of fecal samples was further analyzed using LEfSe analysis. The LEfSe analysis showed that *Treponema*, *Ruminococcus*, *Streptococcus*, *Coprococcus*, *Desulfovibrio*, *Akkermansia*, and *Bacillus* were enriched in GES30 sows; *Ruminococcaceae*, *Oscillospira*, *Lactobacillus*, and *CF231* were enriched in GES109 sows; the phyla *Proteobacteria*, *Fusobacteriia*, and *Actinobacteria* and the genera *Bacteroides*, *Clostridium*, *Escherichia*, *Oribacterium*, *Fusobacterium*, and *Blautia* were enriched in LAC3 (Figure 3A). Furthermore, on GES109, the phylum *Firmicutes* and family *Ruminococcaceae* were enriched in SF-fed sows and the phylum *Bacteroidetes* and five genera (*Bacteroides*, *Blautia*, *Clostridium*, *Succinivibrio*, and *Parabacteroides*) were significantly enriched in CON-fed sows (Figure 3B). On LAC3, the phylum *Firmicutes* and three genera (*Oscillospira*, *YRC22*, and *CF231*) were remarkably enriched in SF-fed sows, while the phylum *Proteobacteria* and three genera (*Adlercreutzia*, *Sarcina*, and *Escherichia*) were significantly enriched in CON-fed sows (Figure 3C). Thus, the above data suggest that the microbial composition of sows at perinatal period were greatly changed by dietary SF during pregnancy.

### 2.2. Dietary Soluble Fiber Increases the Intestinal Propionate Production and Plasma OCFAs Concentrations of Breeding Sows

Considering that species belonging to *Eubacterium* (e.g., *Eubacterium hallii*) were demonstrated to contribute to propionate formation [26,27], we further investigated whether the SF supplementation during pregnancy changes intestinal propionate production. As shown in Figure 4, SF addition remarkably increased the levels of fecal or plasma propionate before and after meal (Figure 4A–C). Moreover, breeding stage had significant effects on levels of plasma propionate of sows with data on GES109 and LAC3 being much higher than GES30, regardless of meal (Figure 4B,C). Additionally, fecal propionate of sows in GES109 was higher than that in GES30 and LAC3 (Figure 4A). There was no interaction effect between breeding stage and SF on intestinal propionate production was observed.

In rodent model and humans, odd-chain fatty acids (OCFAs) can be synthesized endogenously from gut-derived propionate or dietary fiber supplementation [21,28]. Thus, we next detected the effects of SF supplementation on plasma non-esterified fatty acids (NEFA) concentrations of sows. As shown in Table 1, dietary SF supplementation remarkably increased the fasting plasma C15:0, C17:0, and OCFAs concentrations and decreased plasma C16:0, C18:0, Even-SFA, and total NEFAs levels. In addition, breeding stage had noteworthy effects on plasma NEFA composition. Briefly, the plasma C16:0, C18:0, C18:1n-9, C20:4n-6, Oven-SFA, and total NEFA levels in GES109 and LAC3 sows were significantly higher than those in GES30 sows (Table 1). Interactive effect between breeding stage and dietary SF on C18:0 was also detected. Taken together, dietary SF addition increases the gut-derived propionate levels and plasma OCFAs concentrations in breeding sows.

### 2.3. Dietary Soluble Fiber Ameliorates Insulin Desensitization and Systemic Low-Grade Inflammation in Sows at Perinatal Period

In breeding sows, recent works suggest that the sows undergo insulin insensitivity and systemic low-grade inflammation during perinatal period [10]. The excessive insulin desensitization and inflammatory response may reduce feed intake of sows during lactation thereby restrict piglet growth and development [3]. Notably, our recent studies showed that maternal SF diet during pregnancy improves growth performance in piglets [29]. Thus, we hypothesized that dietary SF during gestation might improve the metabolic status of sows. We first detected the plasma glucose levels before and after a meal test and a glucose tolerance test. The circulating pro-inflammatory cytokines (TNF-a and IL-6) before morning meal were also assessed. As shown in Figure 5, between 15 and 240 min after meal at GES109, the SF sows had lower plasma glucose levels than those of CON sows (Figure 5A). Furthermore, the area under curve of the plasma glucose concentrations in LAC3 were markedly higher than in GES109, while the inclusion of SF in pregnancy diet greatly reduced the area under curve of plasma glucose (Figure 5C). Moreover, between 20 and 50 min after glucose infusion at GES109, the SF sows had lower plasma glucose levels than those of CON sows (Figure 5D). Additionally, the inclusion of SF in pregnancy diet significantly reduced the area under curve of plasma glucose levels at GES109 (Figure 5F). Although there was no significant difference in plasma TNF-a levels among different breeding stages (Figure 6B), the plasma IL-6 levels increased gradually from GES30 to LAC3 (Figure 6A). Notably, the inclusion of SF in pregnancy diet dramatically reduced the levels of plasma IL-6 and TNF-*α* of sows (Figure 6A,B). Considering the OCFAs were shown to be inversely correlated with type 2 diabetes or insulin resistance in humans or rodent models [19,30], a Spearman correlation analysis was performed to test the correlation between plasma C15:0 and C17:0 levels and the AUC of glucose after meal and plasma pro-inflammatory cytokines (TNF-a and IL-6). The results showed that the plasma C15:0 and C17:0 was negatively correlated with the AUC of glucose after meal and plasma IL-6 (Figure 6C). In summary, these results indicate that dietary SF improves insulin sensitivity and alleviates systemic low-grade inflammation in sows at perinatal period, potentially related to its stimulating effect on C15:0 and C17:0 production.

## 3. Discussion

As reported by Koren et al. [10] and Cheng et al. [11], the gut microbiota alteration during pregnancy may be closely related to maternal pregnancy-induced metabolic changes. Therefore, the gut microbiota may be an important regulatory target for improving maternal metabolic health. Although dietary SF has been demonstrated to effectively regulate the gut microbiota structure of non-pregnant individuals [31,32], the impact of dietary SF on gut microbiota in pregnant sows is still largely unclear. In the present study, our data indicate that dietary SF dramatically alter the gut microbiota diversity and composition of breeding sows.

Gut microbiota richness and stability are considered critical for host–microbe symbiosis, because it helps to maintain beneficial symbiotic bacteria and their related functions over time [33,34]. Accordingly, major or frequent changes in microbial structure are often associated with ill health [35,36,37]. The results of alpha and beta diversity analysis showed that the gut microbiota structure of perinatal sows changed remarkably, which is good consistent with the results of our previous study [11]. In that study, we suggested that perinatal microbial richness and diversity reduction are closely linked with sow’s metabolic disorders [11]. Importantly, in the current study, we found that dietary SF during pregnancy can effectively promote the stability of gut microbiota of sows in perinatal period by stabilizing the gut microbiota richness, which is consistent with previous studies conducted on non-pregnant individuals [38,39].

The abundant phyla of gut microbiota in sows were in good agreement with other studies [40,41] on pregnant and lactating sows with *Firmicutes* and *Bacteroidetes* being the most dominant phyla. Nevertheless, it should be pointed out that the dominant genera in this study were *Eubacterium*, *Prevotella*, *Oscillospira*, *Bacteroides*, and *Treponema*, which are somewhat different from the results of other studies on sows [40,42]. For example, the dominant genera in Huanjiang sows are *Lactobacillus*, *Treponema*, *Ruminococcus*, *Clostridium*, and *Prevotella*, while those in Landrace × Yorkshire sows are *Treponema*, *Clostridium*, *Ruminococcus*, *Prevotella*, and *Streptococcus*. This indicates that the genetic background is an important factor shaping the gut microbiota composition of sows. Noteworthily, dietary SF addition remarkably increased the relative abundance of *Firmicutes* in perinatal sows. The LEfSe analysis further revealed that dietary SF supplementation promoted the enrichment of the family *Ruminococcaceae* in late pregnancy and the accumulation of the genus *Oscillospira* in early lactation. This finding is consistent with the result of Zhou et al. [40], who showed that 1.5 % inulin addition improved *Oscillospira* in pregnant sows. The family *Ruminococcaceae* and genus *Oscillospira* have been identified as important butyrate-producing bacteria [43,44]. In another unpublished study, we did confirm that the inclusion of SF in pregnancy diet increased butyrate production, thereby reducing intestinal permeability and local intestinal inflammation in perinatal sows. Therefore, increasing the butyrate-producing bacteria and butyrate production is one of the ways in which dietary fiber during pregnancy improve the gut health of sows. 

Remarkably, an increase of genus *Eubacterium* was observed in SF-fed sows in this study. The *Eubacterium* spp., which is the common genus of adult gut microbiota, plays a crucial role in intestinal metabolic balance due to its ability to produce butyrate from the fermentation intermediates lactate and acetate, and utilizes 1,2-propanediol to form propionate [26,45]. Thus, this data prompted us to evaluate intestinal propionate production in SF-fed sows. Consequently, we found that propionate was increased in the feces and plasma of SF-fed sows. This finding is in good agreement with our previous study [23] in which we found dietary SF increased the level of cecal propionate in rats. Considering recent evidence suggesting that OCFAs can be used as a biomarker for dietary fiber intake since gut-derived propionate being used for the hepatic synthesis of OCFAs [21], we next detected the plasma NEFA concentrations of SF-fed sows. Our results showed that inclusion of SF in the gestation diet increased the levels of the fasting plasma C15:0, C17:0, and OCFAs concentrations. This finding is consistent with the result of Weitkunat et al. [21], who found that inulin supplementation increased OCFAs levels in non-pregnant individuals. Plasma OCFAs concentrations have been reported to be inversely correlated with the risk of insulin resistance or type 2 diabetes [19,20,30]. In the current study, the meal tests and glucose tolerance tests confirmed that insulin resistance in postpartum was higher than that in prepartum, and SF supplementation improved the insulin sensitivity of sows during perinatal period, typically in prepartum. Moreover, the systemic low-grade inflammation in sows at perinatal period was also alleviated. Interestingly, the Spearman correlation analysis showed that the increased plasma C15:0 and C17:0 levels in SF-fed sows negatively correlated with the AUC of glucose after meal and plasma IL-6. Taken together, our results suggest that the increase of genus *Eubacterium after* supplementing dietary SF during pregnancy promotes propionate and OCFAs production, which in turn improves insulin sensitivity and systemic inflammation in perinatal sows. The improvement of insulin sensitivity by oral treatment with *Eubacterium* has been verified in db/db mice [46]. However, whether genus *Eubacterium* directly promotes the improvement of the metabolic status of sows needs further research. In addition, the underlying mechanisms by which OCFAs improve insulin sensitivity and inflammatory status require further exploration.

## 4. Materials and Methods 

All procedures involving animals were approved (15 January 2016) by the Institutional Animal Care and Use Committee of Huazhong Agricultural University (HZAUSW-2016-023).

### 4.1. Animals, Diets, Housing and Sample Collection

A total of 16 multiparous Landrace sows with an average parity of 4.63 ± 0.62 were selected to this study. The sows were randomly assigned to two treatment groups according to their parity and backfat thickness with 8 sows per group. During pregnancy, the sows were fed with two different diets including a control gestation diet and the same control diet supplemented with 2.0 % guar gum (Yunzhou, China) plus pregelatinized waxy maize starch (Hangzhou, China) to replace 2.0 % rice bran meal (SF; 14.3 % guar gum and 85.7 % pregelatinized waxy maize starch). It was ensured that the different gestational diets have the same content for all nutrients other than soluble fiber level. All sows were fed the same amount of feed during the whole gestation. In detail, the whole gestation period was divided into four feeding stages, namely, d0 to d30 of pregnancy, d31 to d80 of pregnancy, d81 to d95 of pregnancy, and d96 of pregnancy to parturition; Correspondingly, the feeding amount was 2.0 kg/d, 2.4 kg/d, 2.6 kg/d, and 3.0 kg/d, respectively. Sows were fed twice per day at 0700 and 1400 h. The pregnant sows were housed individually in gestation stalls (2.2 m × 0.7 m × 1.1 m). On day 100 of pregnancy, sows were moved to individual farrowing pen with crates (2.2 m × 0.7 m), slatted floors, and heat pads for piglets. All sows were allowed to consume the same diets ad libitum in lactation. The detailed ingredients and nutrient contents of experimental diets are shown in Appendix A.

Fasting blood samples (10 mL) from ear vein were collected from each sow per treatment before morning meal on day 30 (GES30) and 109 of gestation (GES109) and on day 3 of lactation (LAC3) for propionate, NEFA, and inflammatory cytokine analysis. 4 h after morning meal, the feeding blood samples were also collected at the same day for propionate analysis. Blood samples were collected in heparinized tubes (5 mL). Plasma samples were then obtained by centrifuging the blood samples at 3000× *g* for 10 min at 4 °C and were stored at −80 °C until analysis. Fresh fecal samples were individually collected in duplicate using two sterile 20 mL centrifuge tubes (without any treatment) from each sow at day 30 and 109 of gestation and at day 3 of lactation. All sows did not have disease and diarrhea before sampling. The fecal samples were transported (the tubes frozen on dry ice) immediately to the laboratory and then stored at −80 °C until analysis. The duplicate fecal samples were analyzed for propionate concentration and microbial composition, respectively.

### 4.2. Surgical Procedure, Meal Test, Glucose Tolerance Test, and Sampling

At day 102 of gestation, an indwelling catheter was implanted in the left external jugular vein using a surgical technique described by Père and Etienne [3] with minor modifications. The catheter was tunneled under the skin, externalized on the dorsal surface of the neck, and stored in a small bag sutured to the skin. The duration of surgery never exceeded 1 h. All sows underwent surgery successfully and returned to normal feed intake after 3 days. Catheters were flushed 4 times weekly with a 10-mL saline solution containing 200 IU of heparin/mL. 

Two tests (a meal test and a glucose tolerance test) were used at day 109 of gestation and day 3 of lactation, respectively. The two tests were applied during successive days at each stage and began in the morning after an overnight fasting period of 16 h. Meal tests included measuring plasma glucose and insulin concentrations after ingestion of a meal of 1.2 kg in gestation and 1.5 kg in lactation, respectively. Concentrations were detected in the jugular vein at 30 and 15 min before the meal, at 15-min intervals from 15 to 120 min, and at 30-min intervals from 150 to 240 min after the initiation of the meal (time 0). One day after meal test, glucose tolerance test was carried out by infusing 0.5 g of glucose/kg of BW through the jugular catheter. The first blood sample (time 0) was taken immediately before infusion. Blood samples were also collected 15 and 10 min before the test and at 3-min intervals from 3 to 15 min, at 5-min intervals from 20 to 40 min, and at 10-min intervals from 50 to 60, and 15-min intervals from 75 to 120 min after time zero. All samples were analyzed for plasma glucose and insulin concentrations. 

The same sampling method was applied at each sampling time point. Before a blood sample was taken via the intravenous catheter, 5 mL of blood were withdrawn and discarded to eliminate dilution from the heparin block. 3 mL of blood were then collected in heparinized tubes (5 mL). Plasma samples were then obtained by centrifuging the blood samples at 3000× *g* for 10 min at 4 °C and were stored at −80 °C until analysis. After sampling, a 5-mL saline solution with heparin (20 IU/mL) was injected into the catheters to prevent blood clots.

### 4.3. Analysis of Fecal and Plasma Propionate

The gas chromatography was used to detect the levels of propionate in feces and plasma. The measurement procedure was modified on the basis of the method of Chen et al. [47]. In short, about 1.5 g of feces were first homogenized in 1.5 mL of deionized water. The above homogenate was centrifuged at 11,000× *g* at 4 °C for 10 min, and then collected the fecal supernatant. Thereafter, the fecal supernatant and plasma samples were acidified with 25% metaphosphoric acid at a ratio of 1:5 (1 volume of metaphosphoric acid for 5 volumes of sample). After acidified on ice for 30 min, 1 uL of the supernatant was separated and detected using a gas chromatography (Shimadzu, Japan) equipped with a capillary column (CP-Wax 52 CB, Chrompack, Netherlands). The propionate was quantified using external standard curves from 0.1 to 100 umol/mL of the authentic organic acid (Fluka, Switzerland).

### 4.4. Analysis of Plasma Non-Esterified Fatty Acids

The fasting plasma NEFAs composition was evaluated according to the method described by Eltweri et al. [48] with minor modifications. Briefly, total lipid was extracted from plasma samples with chloroform:methanol (2:1 *v*/*v*). NEFAs were isolated from the plasma lipid extract by solid phase extraction (SPE) on Bond-Elute cartridges. Triacylglycerols and cholesteryl esters were eluted using chloroform while phosphatidylcholine was eluted with chloroform:methanol (60:40, *v*/*v*) and discarded. Thereafter, NEFAs were finally eluted with chloroform:methanol:glacial acetic acid (100:2:2, *v*/*v*/*v*) under vacuum suction. Then fatty acid methyl esters (FAMEs) were formed by reaction with methanol containing 10% (*v*/*v*) sulphuric acid and heating at 62 °C for two hours. After cooling and neutralization with KHCO3, FAMEs were extracted into Isooctane.

FAMEs were separated and determined with a Varian GC 2010 Gas Chromatograph (Shimadzu, Tokyo, Japan) equipped with a CP-Wax 52 CB column 30.0 m × 0.53 mm i.d (Chrompack, Middleburg, Netherlands). The inlet temperature was 260 °C. Nitrogen was used as the carrier gas. FAMEs were detected by using a flame ionization detector held at a temperature of 260 °C. The instrument was controlled by, and data collected, with the HPChemStation software (Hewlett-Packard, Palo Alto, CA, USA). FAMEs were identified and quantified by comparison of retention times with the respective authentic standards run previously. The following nine individual FAs were identified: myristic acid (C14:0), pentadecanoic acid (C15:0), palmitic acid (C16:0), heptadecanoic acid (C17:0), stearic acid (C18:0), oleic acid (C18:1n-9), linoleic acid (C18:2n-6), arachidonic acid (C20:4n-6) and docosahexaenoic acid (C22: 6n-3). Even-chain SFA (Even-SFA) was the sum of C14:0, C16:0 and C18:0 while odd-chain SFA (OCFAs) was the sum of C15:0 and C17:0.

### 4.5. Plasma Glucose and Inflammatory Cytokine Analysis

Plasma glucose was determined with a glucose dehydrogenase activity colorimetric assay kit (BioVision Inc., San Francisco, CA, USA). Plasma concentrations of interleukin-6 (IL-6) and tumor necrosis factor alpha (TNF-*α*) were measured using commercially available porcine ELISA kits (Bio-Swamp, Wuhan, China) according to the manufacturer’s instructions. All samples were analyzed in duplicate.

### 4.6. Microbiota Analysis Based on 16s rRNA High-Throughput Sequencing

Microbial DNA of fecal samples was extracted using the QIAamp Fast DNA Stool Mini Kit (Qiagen, Hilden, Germany) according to manufacturer’s protocols. The V3–V4 hypervariable region of an 16S rRNA gene was amplified using PCR bar-coded primers 341F (5′-ACTCCTACGGGAGGCAGCAG-3′) and 806R (5′-GGACTACHVGGGTWTCTAAT-3′). The PCR was conducted according to our previous method (10). After qualification and purification, the PCR amplicons were paired-end sequenced (2 × 250) with a MiSeq platform (Illumina, San Diego, CA, USA) at the Beijing Genomics Institute (BGI, Beijing, China). The high-quality sequences were clustered to operational taxonomic units (OTUs) at 97% similarity using USEARCH [49]. The taxonomy of each OTU was conducted with Ribosomal Database Project (RDP) classifer program against the Greengenes 16S rRNA database [50,51]. The alpha diversity of the samples was calculated with Observed species, Chao 1, and Shannon index. The principal coordinate analysis (PCoA) based on the weighted UniFrac distance were used to summarize the beta diversity. The linear discriminant analysis coupled with effect size (LEfSe) was used to identify the key bacterial taxa between different treatments [52].

### 4.7. Statistical Analyses

Statistical analysis was performed using the Statistical Analysis System (version 9.4; SAS Institute, Cary, NC, USA). Data were analyzed using the MIXED procedures of SAS according to the following equation: *Y_ij_ =µ + α_i_ + β_j_ + (αβ)_ij_ + t_l_ + ε_ijl_*, where Y*_ij_* is the response variable, *µ* is the overall mean, *α_i_* is the fixed effect of dietary treatments (*i* = CON, SF), *β_j_* is the fixed effect of breeding stages (*j* = G30, G109, and L3), (*αβ*)*_ij_* is the interaction effect between dietary treatment and breeding stage, t*_l_* is the random effect of sows used to explain repeated measurements within individual sow, and *ε_ijl_* is the residual error. A *p* < 0.05 was used to indicate significance between different means. The least significant difference method was used to compare the means when significant interaction effects or main effects occurred.

## 5. Conclusions

In conclusion, the present study suggests that the supplementation of pregnancy diet with SF effectively enhances the stability of gut microbiota structure and greatly changes the composition of gut microbiota in sows. The representative changes in the composition of gut microbiota include a decrease in *Proteobacteria* and an increase in *Ruminococcaceae*, *Oscillospira*, and *Eubacterium*. Moreover, the increase of genus *Eubacterium* after dietary SF supplementation during pregnancy promotes propionate and OCFAs production, which may be one of the potential mechanisms by which dietary SF improves insulin sensitivity and systemic inflammation in perinatal sows. Further studies are needed to elucidate the mechanisms by which OCFAs improve the metabolic status of sows.

## Figures and Tables

**Figure 1 ijms-21-00635-f001:**
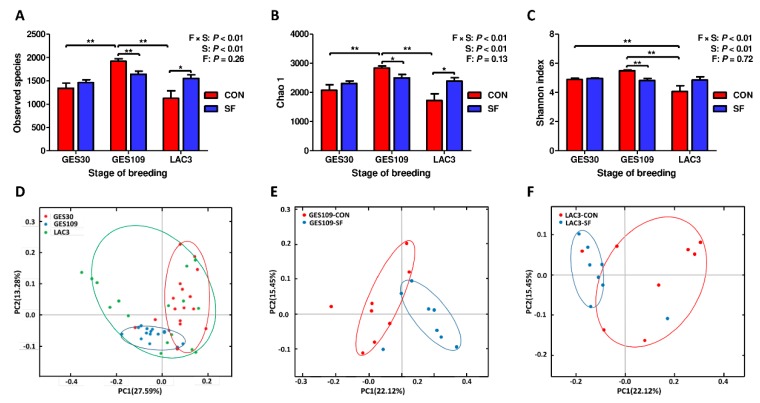
Effects of soluble fiber during gestation on the gut microbiota diversity of sows. (**A** to **C**) Comparison of the number of observed species (**A**), Chao 1 (**B**), and Shannon index (**C**) between CON- and SF-fed sows. (D–F) Beta diversity of gut microbiota in sows among different breeding stages (**D**) and between different dietary treatments at day 109 of gestation (**E**) and day 3 of lactation (**F**) based on weighted UniFrac distance. Data are expressed as mean ± SEM (*n* = 8). When significant interactive effects or main effects were observed, the means were compared using the least significant difference method with a *p* < 0.05 indicating significance. **p* < 0.05; ***p* < 0.01. GES109, day 109 of gestation; LAC3, day 3 of lactation. CON, control diet; SF, 2.0 % soluble fiber combination diet.

**Figure 2 ijms-21-00635-f002:**
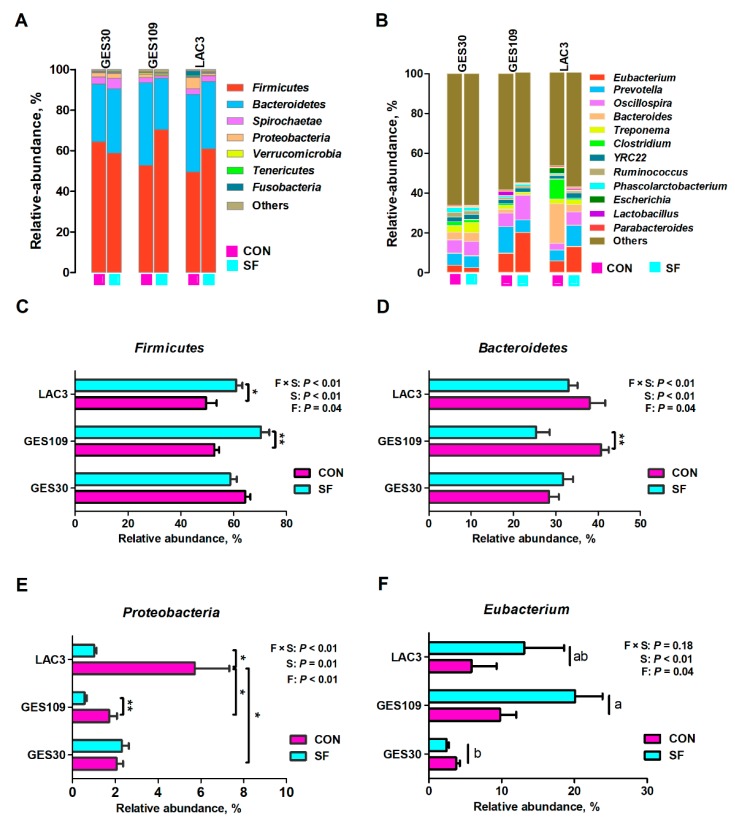
Effects of soluble fiber during gestation on gut microbiota composition of sows. The relative abundance of different phyla (**A**) and genera (**B**) in gut microbiota of CON and SF sows among different breeding stages. Comparison of phyla Firmicutes (**C**), Bacteroidetes (**D**), and Proteobacteria (**E**) and genus Eubacterium (**F**) between CON and SF sows during different breeding stages. Data are expressed as mean ± SEM (*n* = 8). When significant main effects or interactive effects were observed, the means were compared using the least significant difference method with a *p* < 0.05 indicating significance. **p* < 0.05; ***p* < 0.01. a-b, significant effect of breeding stage (*p* < 0.05; values with different lowercase letters are significantly different). GES109, day 109 of gestation; LAC3, day 3 of lactation. CON, control diet; SF, 2.0% soluble fiber combination diet.

**Figure 3 ijms-21-00635-f003:**
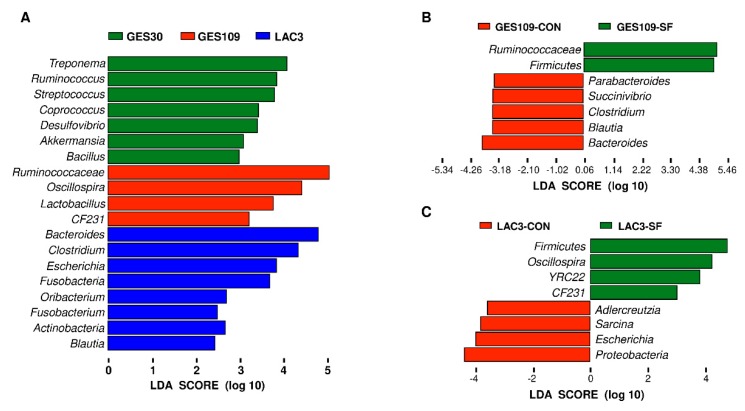
The linear discriminant analysis coupled with effect size (LEfSe) analysis of gut microbiota composition of CON and SF sows. (A–C) The key taxa among different breeding stages (**A**) and between different dietary treatments at day 109 of gestation (**B**) and day 3 of lactation (**C**). GES109, day 109 of gestation; LAC3, day 3 of lactation. LDA, linear discriminant analysis. CON, control diet; SF, 2.0% soluble fiber combination diet.

**Figure 4 ijms-21-00635-f004:**
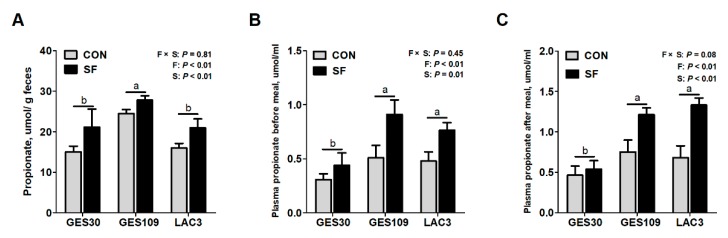
Effects of soluble fiber during gestation on propionate production of sows. (**A**) Fecal propionate. (**B**) Plasma propionate before meal. (**C**) Plasma propionate after meal. Data are expressed as mean ± SEM (*n* = 8). When significant main effects or interactive effects were observed, the means were compared using the least significant difference method with a *p* < 0.05 indicating significance. a-b, significant effect of breeding stage (*p* < 0.05; values with different lowercase letters are significantly different). GES109, day 109 of gestation; LAC3, day 3 of lactation. CON, control diet; SF, 2.0% soluble fiber combination diet.

**Figure 5 ijms-21-00635-f005:**
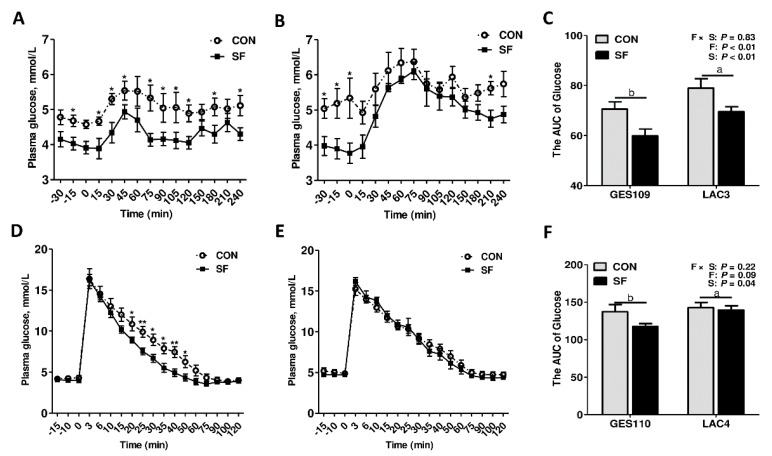
Effects of soluble fiber during gestation on insulin sensitivity of sows. (**A**–**B**) The plasma glucose concentrations of sows after meal test at day 109 of gestation (GES109) (**A**) and day 3 of lactation (**B**). (**C**) The area under curve of plasma glucose after meal tests at day 109 of gestation and day 3 of lactation. (**D**–**E**) The plasma glucose concentrations of sows after glucose tolerance tests at day 109 of gestation (**D**) and day 3 of lactation (**E**). (**F**) The area under curve of plasma glucose after glucose tolerance tests at day 109 of gestation and day 3 of lactation. Data are expressed as mean ± SEM (*n* = 8). **p* < 0.05; ***p* < 0.01. a-b, significant effect of breeding stage (*p* < 0.05; values with different lowercase letters are significantly different). GES109, day 109 of gestation; LAC3, day 3 of lactation. CON, control diet; SF, 2.0% soluble fiber combination diet.

**Figure 6 ijms-21-00635-f006:**
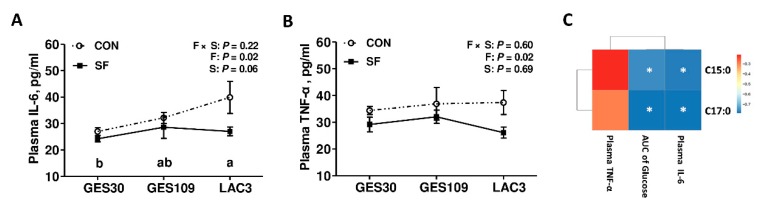
Effects of soluble fiber during gestation on systemic inflammation of sows. Comparison of plasma IL-6 (**A**) and TNF-*α* (**B**) between CON- and SF-fed sows. (**C**) Heatmap of the Spearman R correlations between odd-chain fatty acids and insulin sensitivity and systemic inflammation parameters of sows. Data are expressed as mean ± SEM (*n* = 8). **p* < 0.05. a-b, significant effect of breeding stage (*p* < 0.05; values with different lowercase letters are significantly different). GES109, day 109 of gestation; LAC3, day 3 of lactation. CON, control diet; SF, 2.0% soluble fiber combination diet. IL-6, interleukin 6; TNF-*α*, tumor necrosis factor *α*.

**Table 1 ijms-21-00635-t001:** Effects of soluble fiber supplementation to gestational diet on fasting plasma concentrations of non-esterified fatty acids of sows^1,2^.

Item	Soluble Fiber Level		Reproductive Stage			SEM	*p*-Value		
	0	2%	GES30	GES109	LAC3		F	S	F × S
Plasma NEFA before meal, µmol/L									
**C14:0**	1.53	0.97	0.59 ^b^	1.03 ^b^	1.87 ^a^	0.18	0.12	< 0.01	0.24
**C15:0**	1.34 ^B^	2.34 ^A^	1.91	2.12	1.63	0.21	0.02	0.58	0.66
**C16:0**	123.46 ^A^	94.91 ^B^	53.67 ^b^	133.78 ^a^	128.32 ^a^	7.80	0.02	< 0.01	0.33
**C17:0**	1.66 ^B^	3.26 ^A^	1.57	2.51	3.18	0.32	0.01	0.09	0.22
**C18:0**	68.37 ^A^	45.98 ^B^	32.07 ^b^	65.25 ^a^	67.63 ^a^	4.84	< 0.01	< 0.01	0.02
**C18:1n-9**	79.32	64.88	30.98 ^b^	93.19 ^a^	84.41 ^a^	6.08	0.17	< 0.01	0.91
**C18:2n-6**	172.14	150.03	64.95 ^c^	216.04 ^a^	186.15 ^b^	11.67	0.13	< 0.01	0.96
**C20:4n-6**	85.21	68.29	51.32 ^b^	82.82 ^a^	89.41 ^a^	5.37	0.11	< 0.01	0.99
**C22: 6n-3**	0.78	0.48	0.23	1.01	0.60	0.14	0.23	0.07	0.09
**Even-SFA**	193.36 ^A^	141.86 ^B^	86.33 ^b^	200.06 ^a^	197.82 ^a^	12.16	< 0.01	< 0.01	0.08
**OCFAs**	3.00 ^B^	5.60 ^A^	3.48	4.63	4.81	0.45	< 0.01	0.39	0.54
**Total-NEFA**	533.80 ^A^	431.14 ^B^	237.29 ^b^	597.75 ^a^	563.20 ^a^	31.08	0.01	< 0.01	0.55

^A,B^ Significant effect of dietary treatment (*p* < 0.05; values with different uppercase letters are significantly different); ^a–c^ significant effect of breeding stage (*p* < 0.05; values with different lowercase letters are significantly different); ^1^ Data are expressed as mean ± largest SEM (*n* = 8); ^2^ GES109, day 109 of gestation; LAC3, day 3 of lactation. CON, control diet; SF, 2.0% soluble fiber combination diet. NEFA, non-esterified fatty acids; Even-SFA is the sum of C14:0, C16:0 and C18:0; OCFAs is the sum of C15:0 and C17:0; Total-NEFA is the sum of nine individual fatty acids.

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
