# Peer review of "Inclusion of Soluble Fiber in the Gestation Diet Changes the Gut Microbiota, Affects Plasma Propionate and Odd-Chain Fatty Acids Levels, and Improves Insulin Sensitivity in Sows"

_ijms, 2020, doi:10.3390/ijms21020635_

Round 1

Reviewer 1 Report

This is a high-quality study and it is recommended for publication. The following minor changes are recommended:

Need to describe the housing conditions of the sows.

P. 1, Line 23: what are “positive” microbial changes? You may define or modify the use of word positive.

P. 2, Line 80: “Figure 1A” seems to be a typo.

p.6, Lines 177-178: Clarify the scenario by adding “fasting” in the title of Table 1.

P. 8, Line 273: is “in” good agreement with…

P. 9, Line 303, what is GP?

Author Response

RE: Inclusion of Soluble Fiber in the Gestation Diet Changes the Gut Microbiota, Affects Plasma Propionate and Odd-chain Fatty Acids Levels, and Improves Insulin Sensitivity in Sows

Dear reviewer,

Thank you very much for your warm composition comments concerning our manuscript entitled “Inclusion of Soluble Fiber in the Gestation Diet Changes the Gut Microbiota, Affects Plasma Propionate and Odd-chain Fatty Acids Levels, and Improves Insulin Sensitivity in Sows” (MS ID: IJMS-684526).

Here we submit a new version of our manuscript, which has been modified according to the comments. Detailed point-by-point responses are provided below. We mark all the changes by using red font in the revised manuscript.

We want to do our best to meet the requirements. If you have any question about this paper, please don’t hesitate to let us know.

Sincerely,

Chuanhui Xu and Chuanshang Cheng

Comment 1: Need to describe the housing conditions of the sows.

Response: Thanks for your comment. We have supplied the description about the housing conditions of the sows. Please see lines 310-312 of the revised manuscript.

Comment 2: Line 23: what are “positive” microbial changes? You may define or modify the use of word positive.

Response: Thanks for your suggestion. We have modify the word “positive” with “Genus Eubacterium” for more specific description to avoid ambiguity. Please see line 23 of the revised manuscript.

Comment 3: Line 80: “Figure 1A” seems to be a typo.

Response: Thanks for your comment. “Figure 1A” is a typo, we have corrected to “Figure 1”. Please see line 80 of the revised manuscript.

Comment 4: Lines 177-178: Clarify the scenario by adding “fasting” in the title of Table 1.

Response: Thanks for your suggestion. We have revised it accordingly. Please see lines 177-178 of the revised manuscript.

Comment 4: Line 273: is “in” good agreement with…

Response: Thanks for your comment. We have revised it accordingly.

Comment 4: Line 303, what is GP?

Response: Thanks for your comment. “GP” is a mistake, we have corrected to “SF”. Please see line 303 of the revised manuscript.

Reviewer 2 Report

This study examined the effect of fiber supplementation on microbiome alterations, odd chain fatty acid production, and glucose control in perinatal sows. Results show that fiber supplementation increased diversity of microbiome, increased propionate and odd-chain fatty acid production, and improved markers of glucose control and inflammation. The study was well-designed and the presentation was clear. A few questions are listed below:

Figure 5. Were glucose levels and glucose tolerance measured at GES30? Was there any difference at that “baseline”?

Also, while before meal glucose were higher in control versus supplemented sows as shown in Figure 5A and B, there were no differences in fasting glucose as shown in Figure 5D and E, please discuss this discrepancy.

Since the AUC did not differ between groups at LAC3, it may be better not to state in general that glucose tolerance was improved during the perinatal period by the treatment in Discussion, or otherwise, provide a discussion on this time-point specific difference.

Was there any measurement on the piglets, e.g. birth weight, growth, and adiposity, etc? Any measurement of milk production and composition of the sows? Were there measurements at a later time point of lactation? Including the rationale of time point selection may be helpful.

Minor:

Line 15: “characterized by”, “by” is missing

Line 39: “affect” should be “affects”

Author Response

RE: Inclusion of Soluble Fiber in the Gestation Diet Changes the Gut Microbiota, Affects Plasma Propionate and Odd-chain Fatty Acids Levels, and Improves Insulin Sensitivity in Sows

Dear reviewer,

Thank you very much for your warm composition comments concerning our manuscript entitled “Inclusion of Soluble Fiber in the Gestation Diet Changes the Gut Microbiota, Affects Plasma Propionate and Odd-chain Fatty Acids Levels, and Improves Insulin Sensitivity in Sows” (MS ID: IJMS-684526).

Here we submit a new version of our manuscript, which has been modified according to the comments. Detailed point-by-point responses are provided below. We mark all the changes by using red font in the revised manuscript.

We want to do our best to meet the requirements. If you have any question about this paper, please don’t hesitate to let us know.

Sincerely,

Chuanhui Xu and Chuanshang Cheng

Comment 1: Figure 5. Were glucose levels and glucose tolerance measured at GES30? Was there any difference at that “baseline”?

Response: Thanks for your comments. It was a pity that we did not measure the glucose tolerance at GES30. Limiting to the difficulty of maintaining indwelling catheter for long time, the indwelling catheter was implanted in the left external jugular vein in late gestation (day 102 of gestation) for perinatal measurement of glucose tolerance. In our previous study, we found that the status of insulin sensitivity in sows at GES30 was stable and sensitive [1]. Moreover, dietary supplementation of soluble fiber did not affect the insulin sensitivity of sows in early gestation, which was assessment by homeostatic model assessment [2].

Comment 2: while before meal glucose were higher in control versus supplemented sows as shown in Figure 5A and B, there were no differences in fasting glucose as shown in Figure 5D and E, please discuss this discrepancy.

Response: Thanks for your comment. In the present study, the glucose tolerance tests were performed one day after the meal tests. In meal test, 1.2 kg diet in gestation and 1.5 kg diet in lactation fed to fasting sows might be shortage for a meal of late pregnant sows and lactating sows, which might influence the self-regulation in vivo of glucose during a 16-h fasting period. The excessive starving state of sows might be a reason why there were no differences in fasting glucose in glucose tolerance tests.

Comment 3: Since the AUC did not differ between groups at LAC3, it may be better not to state in general that glucose tolerance was improved during the perinatal period by the treatment in Discussion, or otherwise, provide a discussion on this time-point specific difference.

Response: Thanks for your comment. In the present study, meal tests showed that SF supplementation significantly improved the glucose tolerance during perinatal period (P < 0.01) (Fig. 5C). Glucose tolerance tests showed that SF supplementation decreased the AUC of glucose in glucose tolerance test at GES30, but not the AUC at LAC3 (Fig. 5F), which might be related to the increased glucose intolerance in postpartum compared to that in prepartum. So, our conclusion is adjusted to that SF supplementation improved the insulin sensitivity of sows during perinatal period, typically in prepartum. We have revised it accordingly, please see lines 282-284 of the revised manuscript.

Comment 4: Was there any measurement on the piglets, e.g. birth weight, growth, and adiposity, etc? Any measurement of milk production and composition of the sows?

Response: Thanks for your good suggestions. We have taken note of the changes in piglets. We found SF supplementation improved growth rate and decreased diarrhea incidence of piglets in lactation, and reduced intestinal permeability in piglets [3]. Furthermore, we collected colostrum and normal milk at days 0 (day of delivery), 7,14, and 21 of lactation for analyzing nutritional compositions. Results showed that dietary SF had no significantly effect on milk protein, lactose and milk fat (data not shown).

Comment 5: Were there measurements at a later time point of lactation? Including the rationale of time point selection may be helpful.

Response: Thanks for your comment. Limiting to the difficulty of maintaining indwelling catheter for long time, there were no measurements for glucose tolerance of sows at a later time point of lactation by meal test and glucose tolerance test. Nevertheless, we assessed the effect of SF on glucose tolerance in late lactation (day 14 of lactation) by HOMA model. We found that SF hardly effected the HOMA-IR and HOMA-IS in late lactation (data not shown).

Comment 6: Line 15: characterized by”, “by” is missing.

Response: Thanks for your comment. We have revised it accordingly.

Comment 7: Line 39: “affect” should be “affects”.

Response: Thanks for your comment. We have revised it accordingly.

Cheng, C.; Wei, H.; Xu, C.; Jiang, S.; Peng, J. Metabolic syndrome during perinatal period in sows and the link with gut microbiota and metabolites. Front. microbiol. 2018, 9, 1989. Tan, C.; Wei, H.; Ao, J.; Long, G.; Peng, J. Inclusion of konjac flour in the gestation diet changes the gut microbiota, alleviates oxidative stress, and improves insulin sensitivity in sows. Appl. Environ. Microbiol. 2016, 82, 5899-5909. Cheng, C.; Wei, H.; Xu, C.; Xie, X.; Jiang, S.; Peng, J. Maternal soluble fiber diet during pregnancy changes the intestinal microbiota, improves growth performance, and reduces intestinal permeability in piglets. Appl. Environ. Microbiol. 2018, 84, e01047-01018.
